# SPCBIG-EC: A Robust Serial Hybrid Model for Smart Contract Vulnerability Detection

**DOI:** 10.3390/s22124621

**Published:** 2022-06-19

**Authors:** Lejun Zhang, Yuan Li, Tianxing Jin, Weizheng Wang, Zilong Jin, Chunhui Zhao, Zhennao Cai, Huiling Chen

**Affiliations:** 1College of Information Engineering, Yangzhou University, Yangzhou 225127, China; mx120200526@yzu.edu.cn; 2Research and Development Center for E-Learning, Ministry of Education, Beijing 100039, China; 3Cyberspace Institute Advanced Technology, Guangzhou University, Guangzhou 510006, China; 4Yangzhou Marine Electronic Instrument Research Institute, Yangzhou 225001, China; yz723@vip163.com; 5Computer Science Department, City University of Hong Kong, Hong Kong; weizheng.wang@ieee.org; 6School of Computer and Software, Nanjing University of Information Science and Technology, Nanjing 210004, China; zljin@nuist.edu.cn; 7College of Information and Communication Engineering, Harbin Engineering University, Harbin 150001, China; zhaochunhui@hrbeu.edu.cn; 8Department of Computer Science and Artificial Intelligence, Wenzhou University, Wenzhou 325035, China; cznao@wzu.edu.cn

**Keywords:** blockchain, IoT, smart contract, vulnerability detection, deep learning, serial hybrid network

## Abstract

With countless devices connected to the Internet of Things, trust mechanisms are especially important. IoT devices are more deeply embedded in the privacy of people’s lives, and their security issues cannot be ignored. Smart contracts backed by blockchain technology have the potential to solve these problems. Therefore, the security of smart contracts cannot be ignored. We propose a flexible and systematic hybrid model, which we call the Serial-Parallel Convolutional Bidirectional Gated Recurrent Network Model incorporating Ensemble Classifiers (SPCBIG-EC). The model showed excellent performance benefits in smart contract vulnerability detection. In addition, we propose a serial-parallel convolution (SPCNN) suitable for our hybrid model. It can extract features from the input sequence for multivariate combinations while retaining temporal structure and location information. The Ensemble Classifier is used in the classification phase of the model to enhance its robustness. In addition, we focused on six typical smart contract vulnerabilities and constructed two datasets, CESC and UCESC, for multi-task vulnerability detection in our experiments. Numerous experiments showed that SPCBIG-EC is better than most existing methods. It is worth mentioning that SPCBIG-EC can achieve F1-scores of 96.74%, 91.62%, and 95.00% for reentrancy, timestamp dependency, and infinite loop vulnerability detection.

## 1. Introduction

In recent years, with the progressive development of the pandemic and the rapid development of e-commerce network platforms, the Internet of Things (IoT) devices are more integrated into the privacy of people’s lives. The IoT has encountered many industrial pain points during its long-term development and evolution [1]. “One of the biggest issues in IoT is knowing who you are connecting to. That requirement for trust mechanisms across millions or billions of sensors is what makes a distributed system like a blockchain vital”, said Richard Mark Soley, Ph.D., executive director of the Industrial Internet Consortium based in Needham, Mass. The traditional centralized management approach is difficult to implement effectively in the IoT era: the number of edge and end devices in the IoT era is huge; it is difficult for a single central server or cluster to effectively manage such large-scale devices, and the centralized system faces serious performance bottlenecks. Therefore, the most remarkable permanent data preservation and tamper-proof solution, blockchain, was introduced [2].

In 2017, China Unicom joined many companies and research institutions to set up the world’s first Blockchain of Things (BoT) standard project. This project defines a decentralized platform for trusted Internet services and shows the promise of BoT. Blockchain has an important impact on IoT characteristics, such as subject-to-subject, openness and transparency, secure communication, difficulty being tampered with, and multi-party consensus. The IoT involves devices and sensors that can automatically communicate and share data, and the combination of IoT with smart contracts and blockchain technology is seen as a key innovation in trade finance [3]. Smart contracts allow us to automate complex multi-step processes. The devices in the IoT ecosystem are the points of contact with the physical world [4]. However, the combination of IoT and smart contracts is indeed solving some contradictions and problems [5,6]. For example, there are complex legal issues, as well as security issues such as unpredictable code and contractual vulnerabilities. In the context of the big data era, people’s personalized needs have been met to a greater extent, while at the same time, personal privacy data are also stored in large databases, which also forms a huge hidden risk for network security. There are various means to protect information security. The fundamental task of information network transmission is to protect information from the source to the receiver, thereby sending information securely and preventing it from being damaged, leaked, stolen, tampered with, etc. Safeguarding information security is of great significance to personal information security, commercial secret protection, and government policy operation. Existing studies have secured information security through threat model analysis and password authentication and have achieved significant results [7,8,9]. Whether security can be guaranteed has become one of the most important concerns in various fields in all industries. While BoT technology is vigorously promoting the development of industrial innovation, its security must also be guaranteed. Thus, it is urgent to propose a more efficient approach to smart contract security detection. Our research aims to propose an efficient model with robustness for smart contract vulnerability detection.

The popularity of blockchain technology applications is generating more and more available training data. Designing detection systems that are more sensitive to smart contract security vulnerabilities is the focus of research on smart contract security issues. Deep learning techniques have unique advantages: (a) Deep network models have powerful feature learning capabilities; (b) The performance of the training model improves as the size of the data grows; (c) Features can be extracted directly from the training data [10]. Therefore, combining deep learning techniques with vulnerability detection makes it easier and more effective to detect vulnerabilities in smart contracts by optimizing them through continuous learning. It has the potential to reduce labor costs and improve the accuracy and precision of detection compared to traditional methods.

Typical deep learning models are convolutional neural networks (CNNs), recurrent neural networks (RNNs), deep belief networks (DBNs), etc. CNNs are good at classification, but the convolution and merging steps may lose information about the order and position of words in an utterance [11]; RNNs are better at extracting contextual information about time series [12,13]; and DBNs are commonly used as probabilistic generative models for unsupervised learning and resemble self-coding machines with a slow learning process [14,15,16]. Both CNNs and RNNs can be used for sequence modeling, but they perform differently. RNNs emphasize the order in the time dimension, and the input order of the sequence affects the output. CNNs obtain the overall information by aggregating local information, and they can extract hierarchical information from the input [17]. Existing smart contract vulnerability detection methods incorporating deep learning typically use RNNs [18]. One reason is their advantage in learning nonlinear features of sequences. However, they also suffer from the difficulty of extracting complex vulnerability features by ignoring the traditional code space. A combination of multiple models or mechanisms can achieve better performance in problem classification tasks compared to general deep learning models [19].

A single neural network is limited in its ability to process information. The training accuracy of the network can be improved by increasing the number of neurons and increasing the degrees of freedom of the network. However, this can lead to difficulties in the convergence of the network during training. Eventually, the accuracy requirement cannot be met. Therefore, we propose a robust serial hybrid model based on deep learning. It is a hybrid model that serializes CNNs with RNNs, named Serial-Parallel Convolutional Bidirectional Gated Recurrent Network Model incorporating Ensemble Classifier (SPCBIG-EC). Based on the above research background, we propose an effective hybrid model based on deep learning to detect smart contract vulnerabilities in Ethereum. The model can detect vulnerabilities quickly and efficiently based on patterns learned from training samples. It is a hybrid model that serializes CNNs with RNNs, which we call the **S**erial-**P**arallel **C**onvolutional **Bi**directional **G**ated Recurrent Network Model incorporating **E**nsemble **C**lassifier (SPCBIG-EC).

Our contributions are as follows:We propose the SPCBIG-EC model, which shows excellent performance advantages in multi-task vulnerability detection. For multi-task vulnerability detection in this study, we performed a vulnerability detection task for six typical existing smart contract vulnerabilities. Details of the specific types of vulnerabilities are given below. Meanwhile, we propose a CNN structure, **S**erial-**P**arallel **CNN** (SPCNN), suitable for this serial hybrid model. The SPCNN structure is used for feature extraction to capture the local and global features of the code sequences. The multi-scale series-parallel convolution structure parallelizes the output of low-level and high-level features. It makes the feature combinations of the extracted word vectors more informative.The SPCBIG-EC model uses an Ensemble Classifier. The Ensemble Classifier combines multiple base classifiers to form a powerful classifier. It improves the robustness of the model by combining the decisions of multiple classifier experts.We collected smart contracts from the Ethereum platform and constructed the CESC dataset and the UCESC dataset. They are used for multiple types of vulnerability detection and hybrid vulnerability detection tasks, separately. The CESC dataset consists of snippets of smart contract code after data preprocessing, and UCESC consists of original smart contract code. In our experiments, we focused on six smart contract vulnerabilities, i.e., reentrancy, timestamp dependency, infinite loop, callstack depth attack, integer overflow, and integer underflow.We compared SPCBIG-EC with 11 advanced vulnerability detection methods. Experimental results show that our model has strong advantages in reentrancy, timestamp dependency, and infinite loop vulnerability detection, with F1-scores of 96.74%, 91.62%, and 95.00%. For the hybrid vulnerability dataset, our model also achieved an accuracy and F1-score of over 85%, outperforming most existing detection tools.

## 2. Related Work

### 2.1. Smart Contract Security

Smart contracts are written in a high-level language, compiled into bytecode, driven by blockchain transactions, and run on a virtual machine using the blockchain as a storage base, all of which are subject to different security threats. Smart contract security incidents have occurred frequently in recent years. In 2016, an attack against DAO contracts resulted in the loss of more than 3,600,000 Ethers, which stemmed from a reentrancy vulnerability introduced in a critical DAO contract [20]. In 2017, the Parity multi-signature wallet vulnerability resulted in over 513,701 Ethers being locked. It led to an ongoing debate about whether Ether needs to be upgraded by way of a hard fork [21]. In 2018, hackers attacked the BEC contract, and an integer overflow security vulnerability caused the price of BEC to drop to almost zero [22]. In 2020, the CertiK security research team identified multiple security vulnerabilities in the SushiSwap project’s smart contract. These vulnerabilities could be exploited by the smart contract owner to allow the owner to perform arbitrary operations, including emptying tokens from the smart contract account without authorization [23].

Smart contracts are accompanied by the discovery of different types of vulnerabilities throughout their lifecycle [24]. There are many types of existing discovered smart contract vulnerabilities. Our research focuses on the existing known vulnerabilities, such as reentrancy, timestamp dependency, infinite loop, callstack depth attack, integer overflow, and integer underflow.

### 2.2. Existing Methods for Detecting Smart Contract Vulnerabilities

There are several traditional methods for detecting vulnerabilities in smart contracts. Oyente is a vulnerability detection tool based on static symbolic execution. Oyente’s design is modular, which allows advanced users to implement and insert their detection logic to check custom properties in the contract [25]. Mythril is a vulnerability detection tool based on a combination of symbolic execution and concrete execution, which covers most types of vulnerabilities [26]. In addition, SmartCheck [27], a scalable static analysis tool, and Securify [28], a tool that provides scalable and fully automated security analysis for smart contracts, are advanced tools for smart contract security detection. These automated auditing methods are still in the developmental stage, and they face three main major problems. First, the level of automation is low. They need to have constant feedback to audit. Second, they have a high rate of false positives. They have a low level of automation and still require some human involvement. Third, they have a long audit time [29,30]. Our online data research found that Mythril averaged 60 s, Oyente was about 30 s, and Securify was about 20 s [31].

Traditional smart contract vulnerability detection tools mostly rely on fixed detection rules, while vulnerability detection methods that incorporate deep learning techniques avoid this problem. Yu et al. proposed DeeSCVHunter, a deep learning-based framework for the automatic detection of smart contract vulnerabilities, and they proposed the novel notion of Vulnerability Candidate Slice (VCS) to help models capture the key point of vulnerability [32]. The study provided us with research ideas by helping the model capture the key points of vulnerability. Ashizawa et al. proposed Eth2Vec, a machine learning-based static analysis tool for smart contract vulnerability detection. It automatically learns the features of vulnerable smart contract bytecodes through a neural network for natural language processing [33]. Eth2Vec detects vulnerabilities with a high degree of accuracy by implicitly extracting features and combining lexical semantics between contracts, even after rewriting the code. However, the method suffers from the problem of not supporting inter-contract analysis when multiple contracts are interrelated. Huang et al. developed a smart contract vulnerability detection model based on multi-task learning. It improves the detection capability of the model by setting auxiliary tasks to learn more directional vulnerability features [34]. Liu et al. proposed an interpretable way to combine deep learning with expert pattern models for smart contract vulnerability detection. The model can obtain an interpretable panorama of fine-grained details and weight distributions [35]. Wang et al. proposed a new method called AFS (AST Fuse program Slicing) to fuse code feature information [36]. AFS can fuse structured information from the AST with program slicing information to detect vulnerabilities by learning new vulnerability signature information. A study by Liao et al. proposed SoliAudit (Solidity Audit), which uses machine learning and fuzzy testing for smart contract vulnerability assessment [37]. Distinguishing itself from previous studies, SoliAudit can detect vulnerabilities without expert knowledge or predefined patterns. Mi et al. proposed a new framework, VSCL, which uses DNNs based on metric learning for vulnerability detection in smart contracts. In addition, a new feature matrix was generated by CFG extraction to represent the smart contract and encode the operations using Ngram and TFIDF [38]. Zhang et al. proposed a smart contract vulnerability detection method based on information graphs and integrated learning [39]. This paper proposed an ensemble learning (EL)-based contract vulnerability prediction method, which is based on seven different neural networks using contract vulnerability data for contract-level vulnerability detection. The method exhibited a high capability of vulnerability detection. However, it cannot determine the type of vulnerabilities detected. Although these smart contract vulnerability detection methods based on deep learning techniques have shown excellent performance, most of these methods are optimized to improve the processing of smart contracts, and they all essentially use single-network structure models. They do not optimize the detection capability of the model by changing the network structure. Different types of network structures have different focuses when extracting abstract and semantic features, and a single network structure may have the problem of incomplete extraction of key information due to insufficient learning of semantic and syntactic information of smart contracts, so we consider the perspective of network structures to explore whether feature extraction with hybrid network structures will have a good impact on the vulnerability detection performance of smart contracts.

Given the above research background, we have come to the following conclusions. The deep learning-based vulnerability detection approach is driven by data. It allows fine-grained segmentation of smart contract source code files when building a dataset, which is then fed into the model. Therefore, detection methods using deep learning only require the construction of reasonable datasets, which can lead to a tremendous improvement in the code coverage of detection and thus reduce the rate of missed detections. A single neural network is limited in its ability to process information. The training accuracy of the network can be improved by increasing the number of neurons and increasing the degrees of freedom of the network [40]. However, this can lead to difficulties in the convergence of the network during training. Eventually, the accuracy requirement cannot be met. The combination of multiple models or mechanisms can achieve better performance in classification tasks compared to general deep learning models [41]. Based on the above research background, we propose a serial hybrid model combining CNN and RNN for smart contract vulnerability detection.

## 3. Design of the Model

When using deep learning methods for smart contract vulnerability detection, two practical problems need to be solved: (1) constructing a reasonable smart contract source code dataset that can be trained by deep learning models; (2) constructing suitable deep learning models for the smart contract source code dataset. For both problems, we designed the overall implementation of the task, as shown in Figure 1. 

Our work in this study is divided into two parts: processing the data and passing the processed data into a deep learning model to analyze and judge it. It is worth mentioning that existing studies, such as model design and system implementation, fully consider the security issues among modules, and thus, many technology-related studies on authentication have been widely proposed and applied [42,43,44]. The design of our model involves the components of each module, but they are all linear connections of neural network modules, so the security guarantee between each module does not need further consideration here.

The main feature of this paper is the construction of deep learning models, in which we build serial hybrid models combining CNN and RNN. Our approach is motivated by the following considerations:(1)The smart contracts that we process are sequential information. Both CNNs and RNNs can be used for sequence modeling, but they perform differently. RNNs emphasize the order in the time dimension, and the input order of the sequence affects the output. CNNs obtain the overall information by aggregating local information, and they can extract hierarchical information from the input.(2)RNNs read and interpret the input information of a code vector in a single pass, so the deep neural network must wait to process the next code vector until the current information has been processed. This means that RNNs cannot take advantage of massively parallel processing (MPP) as CNNs can [45].(3)CNNs, while achieving good results in feature extraction, do not even consider the contextual relationships of the sequence. The occurrence of each word in the code is considered independent of other words. However, a smart contract code is a long sequence of words, and the occurrences of individual words are contextually interrelated.(4)A single neural network is limited in its ability to process information. The combination of CNN and RNN enables the temporal structure and location information of sequence data to be fully preserved. It facilitates the extraction of multivariate combinatorial features.

### 3.1. SPCBIG-EC Model

We use a Serial-Parallel Convolutional Bidirectional Gated Recurrent Network Model incorporating Ensemble Classifier (SPCBIG-EC) for vulnerability detection. The process is as follows. 

Step 1: The smart contract source code is processed into a dataset that can be used by the model. The processed dataset *D* still consists of some smart contracts and can be represented as $D=\left\{C_{1},C_{2},C_{3},\ldots ,C_{n}\right\}$. A smart contract $C_{i}$ consists of many functions that can be represented as $C_{i}=\left\{f_{i,1},f_{i,2},f_{i,3},\ldots ,f_{i,m}\right\}$, where 1≤i≤n. The function $f_{i,j}$ is composed of the lines of code $\left\{c_{i,j,1},c_{i,j,2},c_{i,j,3},\ldots ,c_{i,j,k}\right\}$. It can be expressed as $f_{i,j}$=$\left\{c_{i,j,1},c_{i,j,2},c_{i,j,3},\ldots ,c_{i,j,k}\right\},\;$where 1≤j≤m.

Step 1.1: Data preprocessing is performed to remove content that is not relevant to the vulnerability.

Step 1.2: The source code is divided into small pieces of smart contract code around key points corresponding to different vulnerabilities. These code fragments are logically executable.

Step 2: Small fragments of smart contract code generated by control flow analysis are converted into vector representations.

Step 2.1: User-defined variables and functions are mapped to symbolic names; the segments in the symbolic representation are divided into a sequence of tokens by lexical analysis. In this case, the original contract fragment line is represented by a sequence of multiple tokens containing keywords, operations with built-in normalization variables, etc. The code line $c_{i,j,w}$ is composed of an ordered set of tokens, $c_{i,j,w}=\left\{t_{i,j,w,1},t_{i,j,w,2},t_{i,j,w,3},\ldots ,t_{i,j,w,q}\right\}$, where 1≤w≤k.

Step 2.2: Contract fragment tokens are converted to vectors via Word2Vec [46,47,48]. Word2Vec maps tokens to integers and then converts them to fixed dimensional vectors. The entire contract fragment is mapped as in step 2.1, concatenating each line of the marker to a long list. Word2Vec creates a vector for each segment and obtains a vector of contract segments by combining the tag embedding. We can obtain the word vector $X_{i}$.

Step 3: The processed vectors are fed into the SPCBIG-EC model that we built. The first layer of the model is the convolution layer, and the convolution kernel convolves the input vectors under a convolution window to obtain the combined features $C_{n}^{1}$. When there are k convolution kernels convolving the word vectors under the convolution window, the combined feature $C_{n}^{1}$ can be expressed as $C_{n}^{1}=\left[C_{1n}^{1},C_{2n}^{1},\ldots ,C_{kn}^{1}\right]$.

Step 4: The second layer of the model is the convolution layer, where the convolution kernel convolves the input vectors under a convolution window to obtain the combined features $C_{n}^{2}$.

Step 5: $C_{n}^{1}$ is concatenated with $C_{n}^{2}$ to form a new feature combination. We use the obtained feature vectors as input to the bidirectional gated recurrent (BiGRU) model.

Step 6: The hidden state of the last moment’s BiGRU [49] is fed into the attention mechanism layer as a feature vector. The attention mechanism assigns weights to the significant words in the code. The output is obtained by multiplying the normalized weights obtained through the attention mechanism with the input feature vector of the layer.

Step 7: The attention mechanism layer is connected to the fully connected layer, which contains an Ensemble Classifier. The Ensemble Classifier contains several weak classifiers as candidate base classifiers (two classifiers, AM-Softmax [50] and Softmax [51,52], were used as examples in our experiments). The weak classifier with the smallest classification error is selected as the base classifier and trained iteratively using the re-weighting method. The base classifiers are combined using a sequential linear weighting approach to obtain a stronger classifier with higher robustness. The constructed strong classifiers are used as the final classifiers to predict vulnerabilities.

Figure 2 shows the structural diagram of our model. The implementation of each module is described in detail in the remaining subsections of Section 3.

As shown in Figure 2, the SPCNN structure consists of multiple convolutional layers, which are based on a two-layer serial convolutional structure. Serial in our structure refers to the linear serial structure of two convolutional layers. Parallel refers to the fact that features extracted from the first convolutional layer are backed up and kept before being passed to the second convolutional layer, and the backed-up features are output in parallel with features extracted from the second convolutional layer from the higher convolutional layer, and the two features are fused to obtain the final extracted feature information of the SPCNN structure. The feature data $C_{n}^{1}$ extracted from the first convolutional layer is kept for backup, while the data are input to the second layer for secondary feature extraction, and the feature extraction data $C_{n}^{1}$ from the first layer under retention is output in parallel with the feature extraction data $C_{n}^{2}$ output from the second layer at the end of the second layer for feature fusion; then, the semantic and syntactic information of the smart contract is fully extracted by taking the bottom layer features and parallelizing the output at the top layer. When there are k convolution kernels convolving the word vectors under the convolution window, the combined feature $C_{n}^{1}$ can be expressed as $C_{n}^{1}=\left[C_{1n}^{1},C_{2n}^{1},\ldots ,C_{kn}^{1}\right]$. When there are s convolution kernels convolving the word vectors under the convolution window, the combined feature $C_{n}^{2}$ can be expressed as $C_{n}^{2}=\left[C_{1n}^{2},C_{2n}^{2},\ldots ,C_{sn}^{2}\right]$. The final features extracted by the SPCNN structure can be expressed as $C=concatenate\left(C_{n}^{1},C_{n}^{2}\right).\;$The advantages of the SPCNN structure are: (1) replacing the single-layer convolution of equivalent steps with a double-layer serial-parallel convolution, which serves to improve the nonlinearity; (2) reducing the impact of the possible loss of key information, such as local position information and destruction of sequence structure, by the convolution pooling operation.

Recurrent neural networks are good at dealing with sequential aspects due to their memory function. With GRU, as a unidirectional recurrent neural network, the next moment’s predicted output is jointly influenced based on the inputs from multiple previous moments. However, the semantic structure among smart contracts is complex, and the prediction results may be influenced by the inputs of both past moments and future moments, so our model considers a bidirectional gated recurrent (BiGRU) network for temporal modeling. BiGRU is a neural network model consisting of unidirectional, oppositely oriented outputs jointly determined by the states of two GRUs. As shown in Figure 2, at each moment, the input will provide two GRUs with opposite directions simultaneously, and the output is jointly determined by these two unidirectional GRUs.

Ensemble learning often achieves significantly better generalization performance than a single learner by combining multiple learners. There are various aggregation methods for ensemble learning, and in this paper, we use the sequence integration method, as shown in Figure 2. It consists of iteratively using weakly learned classifiers and joining their results into a final strongly learned classifier. The joining process usually assigns different weights based on their classification accuracy. After adding a weak learner, the data are usually re-weighted to reinforce the classification of previously misclassified data points. First, we select a classifier from the candidate classifier set as the base classifier and then stack the base classifiers in layers. Each layer is given a higher weight for samples that are wrongly classified by the base classifier in the previous layer during training. When testing, the final result is obtained by weighting the results of each layer of classifiers. This makes use of the dependencies between the base learners. By assigning higher weights to the samples that were incorrectly labeled in the previous training, the overall prediction can be improved.

The implementation principles of each module of the model are described in detail in Section 3.2, Section 3.3, Section 3.4 and Section 3.5.

### 3.2. Dataset Processing

The dataset processing process consists of two steps: (1) converting the smart contract source code into code fragments and (2) labeling the smart contract with vulnerabilities. We describe this process in detail below.

(1)Converting smart contract source code into code fragments.

Lexical analysis of Solidity source code. The main purpose is to convert symbolic representations in Solidity language, such as keywords and operators, to the corresponding tokens. Table 1 shows several examples of token representations.

Determine the key points for different vulnerabilities and segment the smart contract code around the key points. After lexical analysis of the Solidity source code, the source code files need to be fine-grained. We extract the relevant code fragments around a key point and combine them, dividing the entire source code into logically executable code fragments. A code fragment consists of several lines of code that have data dependencies or control dependencies on each other. In this case, the original contract fragment line is represented by a sequence of multiple tokens containing keywords, operations with built-in normalization variables, etc. For the six types of vulnerabilities that we are concerned about, we highlight the key points that correspond to them. These keywords are obtained from the official platform of Ethereum. In Table 2, we introduce the correspondence with the example of reentry vulnerability and its key points. One of the features of Ethereum smart contracts is the ability to call and use code from other external contracts. These contracts typically manipulate Ether, often sending Ether to various external user addresses. This operation of invoking external contracts or sending Ether to external addresses requires contracts to submit external calls. These external calls can be hijacked by an attacker, for example, through a fallback function that forces the contract to execute further code, including a call to itself. This way, the code can be repeated in the contract. In the transfer and call process of smart contracts, users who want their contracts to receive Ether must use *fallback ()*. In addition, *call.value ()* is widely used in the transfer process since it is a transfer function without gas restrictions. These two links give malicious contracts the conditions to launch reentry attacks, so they are used as key points corresponding to reentry vulnerabilities. The rules for determining the key points corresponding to several other vulnerabilities are similar. The reason why we determine key points for different types of vulnerabilities is that training and the learning of large sections of contract source code for vulnerability detection of smart contracts will lead to problems of inadequate feature learning and large training costs caused by redundant information, so we slice and reorganize the original contract code around vulnerability-related keywords to form small pieces of executable code containing key information for model learning and training. Finally, depending on the key points of the different vulnerabilities, we use a program slicing algorithm to transform the original smart contract into a smart contract fragment.

(2)Labeling smart contracts.

The input of the model should contain the smart contract and the corresponding vulnerability label. Considering the large number of smart contracts to be labeled, it is difficult to label vulnerabilities manually, so we use existing security audit tools for vulnerability labeling. Different tools have different criteria for identifying and detecting vulnerabilities. To ensure the accuracy of vulnerability signatures, we chose Oyente, Mythril, and Securify as references for labeling vulnerabilities in smart contracts. In order to balance the conservative and strict detection rules between the different tools, we established the following vulnerability labeling rules. For each smart contract, if at least two tools indicate the existence of a vulnerability, we label the smart contract as “1”; otherwise, it is “0”. Figure 3 represent the vulnerable smart contracts detected by Oyente, Mythril, and Securify, respectively. We finally label the vulnerabilities in smart contracts in regions D, E, F, and G as “1” and the rest of the smart contracts as “0”. The labeled dataset can be represented as D={(x1,y1),(x2,y2),…,(xn,yn)}; each instance point consists of an instance with a label, where x1∈ℝ, yi∈{0,1}.

### 3.3. Code Embedding Layer

The code embedding layer converts the contract code into a vector representation that can be used as input to the neural network. We describe this process in detail below.

(1)Preprocessing data, such as removing blank lines, comments, and special characters, while retaining brackets, operators, etc. The source code is divided into small, logically executable pieces of code by extracting the relevant code pieces around a key point and combining them.(2)Splitting code segments with lexical analysis. Keywords, operators, and delimiters in the code are converted to their corresponding tokens, preserving the semantic order to convert the code segment into a list of tokens.(3)Representing the list of tokens as a vector. There are many traditional word vector representations, such as TF-IDF [53,54], One-Hot [55,56,57], etc. They cannot represent word-to-word information. In this paper, Word Embedding [58] is used to model word vectors by converting contract fragment tokens into vectors via Word2Vec. The network maps tokens to integers and then converts them into fixed dimensional vectors.

In Figure 4, we illustrate this process with a sample piece of reentrant code. Step ① in Figure 4 is a complete section of the source code of the smart contract associated with reentry vulnerability. Step ② represents the code segment where the smart contract source code is preprocessed with data. In this phase, we remove irrelevant information, such as blank lines and comments. At the same time, lines of code related to reentrant vulnerabilities are retained. In step ③, we use line 5 of the code in step ② as an example to split the code and convert it to a token sequence. This is achieved by replacing the user-defined function name with “FUNC1” and the user-defined variable with “VAR1”. If more different user-defined functions and variables exist, the corresponding conversions are “FUNC2, FUNC3, …, FUNCn”, “VAR2, VAR3, …, VARn”. Step ④ transforms the list represented in tokens into a vector representation using Word2Vec as the input to our model.

### 3.4. Feature Extraction Layer

#### 3.4.1. Serial-Parallel Convolutional Layer

Although the convolution layer can significantly reduce the number of connections in the network, the number of neurons in the feature mapping group is not significantly reduced. If the classifier is connected directly after the convolution layer, the input dimension of the classifier is still very high, and it is easy to overfit. To solve this problem, a pooling layer needs to be added after the convolutional layer to reduce the feature dimensionality and avoid overfitting. The pooling layer will serve to reduce the number of features when performing feature selection, and reducing the number of parameters may cause the loss of key information at the same time. We propose a string-parallel convolutional neural network structure (SPCNN). Our SPCNN structure is designed to reduce the impact of the loss of key information, such as local position information, and the damage to the sequence structure caused by the convolutional pooling operation. Additionally, in the process of extracting features, the features within each contract and the relational features between individual contracts are extracted to be adequate. 

SPCNN consists of multiple layers of convolution, and the convolution method used is a one-dimensional wide convolution. A padding of 0 values is used to keep the length of the sequence at the output unchanged after convolution. We denote the dimension of the word vector by d, and s is the size of the sliding window. The convolution kernel is denoted as K∈ℝs×d, where d denotes the length of the convolution kernel, and s denotes the width of the convolution kernel. The k word vectors falling into K can be denoted as xi,xi+1,xi+2,…xi+k−1, which can be represented as a matrix Xi∈ℝs×d.
(1)Xi=[xi,xi+1,xi+2,…,xi+k−1]

The next step is to perform a convolution operation on all of the word vectors above. The multidimensional convolution matrix X is operated by the convolution kernel K∈ℝs×d to generate a new feature map H∈ℝn−k+1. Each feature is denoted by hi, f denotes the nonlinear activation function, bc∈ℝ represents the bias matrix, and i∈n−k+1.
(2)hi=f(K·Xi:i+k,i:s+bc)
(3)H={h1,h2,h3,…,hn−k+1}

Maximum pooling of the feature map, i.e., obtaining only the maximum eigenvalue of each dimension within the pooling window, is performed.
(4)$$p_{j}=\sum_{i=1}^{n+k-1}\max\left(h_{i}\right)$$

The SPCNN uses multiple convolution kernels to form feature maps, and for n feature maps after convolution and maximum pooling, the result is:(5)$$P_{j}=\left[p_{1j},p_{2j},p_{3j},\ldots ,p_{nj}\right]$$

After a theoretical introduction to our proposed SPCNN, the practical structure of the SPCNN is further described next. The hierarchical structure of the SPCNN consists of five parts. The hierarchical structure of the SPCNN is given in Figure 5.

(1)Input layer: the input data are smart contract data that have been lexically analyzed and vectorized.(2)Convolution layer 1: Two sizes of convolution kernels (i.e., $m_{1}$ × $n_{1}$, $m_{2}$ × $n_{2}$) are used to convolve the input data separately. Smart contracts have two scales: intra-contractual and inter-contractual. The $m_{1}$ × $n_{1}$ convolution kernel convolves only on the intra-contract scale so that the internal features of each smart contract can be extracted without destroying inter-contractual information. The $m_{2}$ × $n_{2}$ convolution kernel performs simultaneous convolution within and between contracts, allowing correlation features to be extracted between individual contracts. The number of convolution kernels in the layer is i, the convolution step size is $l_{1}$ in all directions, the activation function is a linear correction unit (ReLu), and the padding parameter is SAME. Concatenate layer 1 splices the features obtained from the two convolutional channels in the corresponding rows, keeping the output sequence of the layer.

(3)Convolutional layer 2: Two sizes of convolution kernels (i.e., $m_{3}$ × $n_{3}$ and $m_{4}$ × $n_{4}$) are each used to convolve the output of splicing layer 1. The number of convolution kernels in the layer is j, the convolution step is $l_{2}$ in all directions, the activation function is also ReLu, and the padding parameter is SAME. Concatenate layer 2 splices the features obtained from the two convolutional channels according to their correspondence, while the output sequence retained by splicing layer 1 is spliced with the output sequence of splicing layer 2.

(4)Two fully connected layers: The number of neurons in fully connected layer 1 is 300. Since the model in this paper performs a binary classification task for smart contract vulnerability detection, there are 2 neurons in fully connected layer 2. In addition, a dropout operation is added between the two fully connected layers to prevent the overfitting of the network.(5)Output layer: The convolved feature matrix is output and used as input to the BiGRU neural network.

#### 3.4.2. Time Sequence Modeling Layer

Based on the internal structure of the GRU, we can calculate the hidden state $h_{t}$ of a single GRU according to the following equation, where $h_{t}$ denotes the state of the data after a reset gate, and $z_{t}$ denotes the state after an update gate.
(6)$$r_{t}=\sigma \left(x_{t}W_{xr}+h_{t-1}W_{hr}+b_{r}\right)$$
(7)$$z_{t}=\sigma \left(x_{t}W_{xz}+h_{t-1}W_{hz}+b_{z}\right)$$
(8)$$\tilde{h_{t}}=\tanh\left(x_{t}W_{xh}+\left(r_{t}⊙h_{t-1}\right)W_{hh}+b_{h}\right)$$
(9)$$h_{t}=z_{t}⊙h_{t-1}+\left(1-z_{t}\right)⊙\tilde{h_{t}}$$

BiGRU is a bidirectional implementation of the GRU network. We denote $\vec{h_{t}}$ as the forward propagated hidden state value and $\overset{\leftarrow }{h_{t}}$ as the backward propagated hidden state value, and then the hidden value of the output at moment t can be expressed by Equation (12). $F_{BiGRU}$ denotes the final feature representation when using BiGRU for time sequence feature extraction.
(10)$$\vec{h_{t}}=GRU\left(x_{t},\vec{h_{t-1}}\right)$$
(11)$$\overset{\leftarrow }{h_{t}}=GRU\left(x_{t},\overset{\leftarrow }{h_{t-1}}\right)$$
(12)$$h_{t}=w_{t}\vec{h_{t}}+v_{t}\overset{\leftarrow }{h_{t}}+b_{t}$$
(13)$$F_{BiGRU}=\left(h_{1},h_{2},h_{3,\ldots ,}h_{t}\right)$$

$w_{t}$ and $v_{t}$ are the weights between the hidden layers in different directions at moment t. To highlight the importance of the impact of different keywords on the reentrancy vulnerability, a self-attention mechanism is introduced after the BiGRU layer of the model to assign weights to different words. The input of the attention mechanism layer is $h_{t}$, which is the hidden vector of the output processed by the BiGRU neural network layer in the previous layer.

First, we input the hidden layer vector to the fully connected layer (with weight W and bias b) to obtain $u_{t}$, which is stated in Equation (14).
(14)$$u_{t}=\tan h\left(Wh_{t}+b\right)$$

Then, we use this vector to calculate the calibration vector $\alpha_{t}$ (weights normalized by the attention mechanism), which is stated in Equation (15).
(15)$$\alpha_{t}=\frac{\exp\left(u^{T}u\right)}{\sum_{t}\left(\exp\left(u^{t}u\right)\right)}$$

$u^{t}$ in the above equation is the average optimal vector corresponding to the current time step *t*, which is different for each time step. This vector is obtained by training. The output of the attention mechanism layer is expressed in Equation (16).
(16)$$s_{t}=\sum_{i=1}^{t}\alpha_{t}h_{i}$$

### 3.5. Classification Optimization Layer

Our ensemble approach is as follows: the feature samples extracted by the deep learning model are split and given initial equal weights, and the existing model is given higher weights by adaptively changing the distribution of the training samples, so training samples with errors in the previous classification model receive more attention in the subsequent one to compensate for the shortcomings of the existing model. By changing the weight distribution of the training data in each iteration, the data play different roles in the learning of each weak classifier, so different data play their own roles, and the classification error rate of the base classifier constructed after each iteration decreases steadily with the increase in the number of iterations, which improves the robustness of the model. Under our integrated classification framework, a variety of regression classification models can be used to build weak learners with great flexibility.

In our model, we construct a set of candidate classifiers. The classifiers in our candidate classifier set are Softmax, AM-softmax, and SVM. We would like to state that there can be multiple single classifiers in the candidate classifier set, and in addition to the classifiers involved in our study, Naive Bayes, KNN, etc., can be a member of the candidate classifier set. Introducing a fairness mechanism for selecting weak classifiers in the integrated classifier, we determine the kind of base classifier by comparing the classification accuracy of single classifiers in the classifier set through one test experiment. The selection of weak classifiers directly determines the performance of base classifiers, and integrated classification algorithms often lack selection criteria for weak classifiers. In order to improve the stability and interference resistance of the model, we add a step of a weak classifier selection process before integration. This method integrates the diversity of classifiers, the accuracy of sub-classifiers, and the accuracy of combined classifiers when searching and selecting weak classifiers, so the classifiers as a whole and locally have high fitness and improve the generalization ability and interference resistance of the model. After determining the base classifier, we use that base classifier for ensemble learning.

In contrast to traditional single classifiers (e.g., Softmax, Decision Trees, Parsimonious Bayes, SVM, etc.), our model uses an integrated classifier in the classification phase, which we call the Ensemble Classifier. We select one classifier from the candidate classifier set as the base classifier. The weights are first updated by continuously learning and optimizing a series of weak classifiers (base classifiers) in the set. These weak classifiers are then combined in a sequential linear weighting fashion to form a strong classifier. This is an incremental process whereby the ith base classifier is based on the i−1th classifier by gradually adding “expert predictions”. The Ensemble Classifier learns a series of weak classifiers by varying the weight distribution of the training data.

In the classification stage, the training data after feature extraction by the neural network is represented as E={(x1,y1),(x2,y2),…,(xn,yn)}, where x1∈ℝ, yi∈{0,1}. First, a single classifier is used as a candidate base classifier for the Ensemble Classifier. Second, the set of classifiers and the weight distribution of the training data are initialized. Finally, the single classifier with the smallest classification error is selected as the base classifier. 

For the selection mechanism of the base classifier, we define the following. Given a single classifier $C_{i}$ and $C_{j}$, let the number of samples in the sample set where $C_{i}$ and $C_{j}$ are correctly classified at the same time be a, the number of samples where $C_{i}$ is correctly and $C_{j}$ is incorrectly classified be *b*, the number of samples where $C_{j}$ is correctly and $C_{i}$ is incorrectly classified be *c*, and the number of samples where both classifiers are incorrectly classified be *d*. Then, the output correlation coefficient of the two classifiers is $\rho_{ij}=\frac{ad-bc}{\sqrt{\left(a+b\right)\left(c+d\right)\left(a+c\right)\left(b+d\right)}}$. The larger the mean value of $\rho_{i}$, the larger the value of $C_{i}$ taken as the base classifier.

Figure 6 shows the learning process of the integrated classifier. The output is a global classification model consisting of multiple basic classification models, each with a certain weight that is used to indicate the confidence of that basic classification model. The final classification result is generated by voting based on the predicted results of the multiple basic classification models multiplied by the weights. The integration steps after determining the base classifier from the candidate classifier set are as follows.

Step 1: Initialize the weight distribution of the training data E.
(17)$$W_{1}=\left(w_{11},w_{12},\ldots w_{1i},\ldots ,w_{1n}\right),\;w_{1i}=\frac{1}{n},i=1,2,\ldots ,n$$

Step 2: Select the base classifier and update weights.

Step 2.1: Train the model using training data with a weight distribution $W_{m}$ to obtain the base classifier, where m=1,2,…,M.
(18)$$G_{m}\left(x\right):X\to \left\{0,1\right\}$$

Step 2.2: Calculate the classification error of $G_{m}\left(x\right)$ on the training dataset.
(19)$$e_{m}=\sum_{i=1}^{n}P\left(G_{m}\left(x_{i}\right)\neq y_{i}\right)=\sum_{i=1}^{n}w_{mi}I\left(G_{m}\left(x_{i}\right)\neq y_{i}\right)$$

Step 2.3: Calculate the coefficients of $G_{m}\left(x\right)$.
(20)$$\alpha_{m}=\frac{1}{2}log\frac{1-e_{m}}{e_{m}}$$

Step 2.4: Update the weight distribution of the training data.
(21)$$W_{m+1}=\left(w_{m+1,1},w_{m+1,2},\ldots w_{m+1,i},\ldots ,w_{m+1,n}\right)$$
(22)$$w_{m+1,i}=\frac{w_{m,i}}{Z_{m}}\exp(-\alpha_{m}y_{i}G_{m}\left(x_{i}\right)),i=1,2,\ldots ,n\;$$
(23)$$Z_{m}=\sum_{i=1}^{n}w_{m,i}\exp\left(-\alpha_{m}y_{i}G_{m}\left(x_{i}\right)\right)$$

Step 3: Construct linear combinations of base classifiers.
(24)$$f\left(x\right)=\sum_{m=1}^{M}\alpha_{m}G_{m}\left(x\right)$$

Step 4: The final Ensemble Classifier is obtained as follows.
(25)$$G\left(x\right)=sign(f\left(x\right))=sign(\sum_{m=1}^{M}\alpha_{m}G_{m}\left(x\right))$$

In our integration algorithm (Algorithm 1), the weights of samples misclassified by the previous round of weak classifiers need to be increased. At the same time, the weights of correctly classified samples are reduced. We take a weighted majority voting approach, increasing the weights of weak classifiers with small classification error rates so that they play a larger role in the voting. Conversely, we reduce the weights of weak classifiers with large classification error rates so that they play a smaller role in voting.
**Algorithm 1** Integration Algorithm to Implement Ensemble Classifier**Input**: E: Training data for classification. **Initialization**: $W_{m}$: the weights of the training data. m=1,2,…,M. a=0.**Output**: G(x): Ensemble Classifier1: **for** i≥0 **and**
i< M **do**2: Calculate $e_{i}$ by Equation (19)3:     $a\;\leftarrow e_{i}$
4:     if $e_{i}\;\leq a$ **then**5:     $a\;\leftarrow e_{i}$
6:     Base ← $G_{i}\left(x\right)$
7:     Calculate the weight of the Base $W_{i+1}$ by Equation (21)8:     i++
9:     **end for**10:   Calculate f(x) by Equation (24)11:   Calculate G(x) by Equation (25)12:   **return** G(x)


Softmax is one of the most common classifiers in deep learning. Although the Softmax classifier is easy to use and effective, it fails to guide the network to learn distinguishing features. In order to efficiently learn features that are compact within classes and discrete between classes, Wang et al. proposed a Softmax (AM-Softmax) classifier with additional edge cosine distances [59].
(26)$$F_{softmax}\left(X\right)=\frac{e^{W_{k}^{T}X}}{\sum_{j=1}^{n_{0}}e^{W_{j}^{T}X}}=\frac{e^{\|W_{k}\|\|X\cos\theta_{k}}}{\sum_{j=1}^{n_{0}}e^{\|W_{j}\|\|X\|\cos\theta_{j}}}$$
(27)$$F_{AM\text{-}Softmax}\left(X\right)=\frac{e^{s\left(\cos\theta_{k}-d_{m}\right)}}{e^{s\left(\cos\theta_{j}-d_{m}\right)}+\sum_{j=1,j\neq k}^{n_{0}}e^{s\cos\theta_{j}}}$$

In the above equation, X denotes the input vector of the fully connected layer, $W_{j}^{T}$ is the fully connected layer weight of the jth output node, is the output value of the corresponding output node when the classification result is k, k $\in \left\{1,2,\ldots ,n_{0}\right\}$, s is the scale scaling factor, and $d_{m}$ is the additional edge cosine distance.

## 4. Experiments and Results

In order to conduct modular comparison experiments, we built a modular serial combination framework during experiments. We call it the SCR Detection Framework. It contains 15 serial combinations of CNN and RNN models, including SPCBIG-EC. By running the SCR detection framework, it can automatically select the best-performing model in a selective series of combinations. The structure is shown in Figure 7 below.

Our experiments were designed to answer the following four Research Questions (RQs):RQ1: Can our SPCBIG-EC model detect multiple kinds of smart contract vulnerabilities, and what is the performance of vulnerability detection?To answer this question, we conducted experiments using the SPCBIG-EC model for the CESC dataset that we constructed. The CESC dataset contains six typical vulnerabilities. We used CESC for multi-task vulnerability detection.RQ2: Did the serial combination of SPCNN and BiGRU in the feature extraction phase make our model more effective? By how much?To answer this question, two sets of experiments were carried out. (a) First, we compared the reentrancy, timestamp dependency, and infinite loop vulnerability detection performance of the SPCNN model, BiGRU model, and SPCBIG-EC model in the CESC dataset. (b) Then, we performed hybrid vulnerability testing for UCESC using our framework to test the performance evaluation results for all combined patterns.RQ3: Can our proposed SPCNN for the vulnerability feature extraction phase of the CNN module make our model more effective? By how much?To answer this question, we used the SCR Detection Framework to evaluate the performance of three models, CNN-BiGRU, SCNN-BiGRU, and SPCBIG-EC, for three typical vulnerabilities.RQ4: How effective is our model when compared with state-of-the-art methods?To answer this question, we compared SPCBIG-EC with state-of-the-art smart contract vulnerability detection methods. Firstly, a comparison was made with existing automated security auditing tools, namely, Oyenete [25], Mythril [26], Smartcheck [27], and Securify [28]. Secondly, we compared it with deep learning-based vulnerability detection methods, namely, DeeSCV [38], Eth2Vec [33], DR-GCN, TMP, GCE, AME [35], and AFS [36].

### 4.1. Dataset

We collected existing smart contracts on Ethereum and constructed two smart contract datasets, represented as CESC and UCESC.

We collected the source code of Solidity smart contracts from the official Ethereum website, with a total of 203,716 smart contracts. It contains 47,398 real and unique sol files consisting of contract code for six possible vulnerability types, namely, reentrancy vulnerability, timestamp dependency, infinite loop, callstack depth attack vulnerability, integer overflow, and integer underflow. The number of sol files containing vulnerabilities is 35,151, and the number of files without vulnerabilities is 12,247. The data show a large difference between the number of contracts that contain vulnerabilities and those that do not. Such a heavily imbalanced dataset would lead to overfitting during training. We needed to balance the dataset before vulnerability detection and classification, balancing positive and negative examples while ensuring an even distribution of each type of vulnerability. Therefore, after obtaining these smart contracts, further sampling and processing were required to collate a dataset that could be used for experiments. We used the synthetic minority oversampling technique (SMOTE) [60,61] to extend the number of minority classes to be close to the number of majority classes. SMOTE is an oversampling technique that interpolates between a small number of classes to generate additional classes. Figure 8 shows the composition of the CESC dataset. The CESC dataset consists of six categorical datasets, the composition of which is shown in Figure 8. We would like to clarify that the reason we distinguished the dataset into UCESC and CESC is that the CESC required for our experiments was obtained after preprocessing UCESC, which is the original smart contract code that has not been preprocessed, and it consists of smart contract code containing six typical vulnerabilities that are not sorted. The UCESC dataset underwent a series of preprocessing efforts, such as data cleaning and tagging lists, to form a CESC dataset that can be used by our model for vulnerability detection. We also conducted direct vulnerability detection experiments on the UCESC dataset to verify the necessity of data cleaning for this study and to demonstrate the efficiency of our proposed structure for different datasets.

### 4.2. Experimental Settings

The experimental model was built on a computer with Intel Core (TM) i7-10875H CPU, NVIDIA GeForce GTX 2060 GPU, and 16 GB RAM.

The whole experiment was divided into a training phase and a testing phase. Our goal in the training phase was to optimize the model parameters by learning the loopholes to obtain a trained model. The testing phase of our work used the test data as input to the trained model, which outputs the prediction results of vulnerability detection. The prediction results were compared with the real tags to measure the performance of our model. In this paper, we use widely used metrics, including accuracy (*ACC*), true positive rate or recall (*TPR*), false positive rate (*FPR*), precision (*PRE*), and F1-score (*F1*). We calculated these values by the following mathematical representations.
(28)$$ACC=\frac{TP+TN}{TP+TN+FP+FN}$$
(29)$$TPR=\frac{TP}{TP+FN}$$
(30)$$FPR=\frac{FP}{FP+TN}$$
(31)$$PRE=\frac{TP}{TP+FP}$$
(32)$$F1=\frac{2\times PRE\times TPR}{PRE+TPR}$$

The concepts represented by the letters involved in the formulas are as follows. True positive (*TP*): the number of samples in which true reentrancy is detected; false positive (*FP*): the number of samples in which false reentrancy is detected; false negative (*FN*): the number of true reentrant samples not detected; true negative (*TN*): the number of false reentrant samples not detected.

In the SCR Detection Framework, we constructed neural network models that combine multiple types of CNN and RNN serial structures as feature extraction modules. We provided three CNN structures, namely, CNN, SCNN, and SPCNN, and five RNN structures, namely, vanilla RNN, GRU, LSTM, BLSTM, and BiGRU.

### 4.3. Experimental Results

(1) Results for Answering RQ1: To determine whether our model can detect multiple types of vulnerabilities, we evaluated the performance of our model on the CESC dataset. The effectiveness is reported in Table 3.

In this set of experiments, we conducted vulnerability detection experiments on six separate datasets from the CESC dataset. According to Table 3, our model can detect these six vulnerabilities to some extent. SPCBIG-EC showed better detection performance when targeting reentrancy, timestamp, infinite loop, and callstack depth attack vulnerabilities. Their accuracy rates are almost all above 90%. In addition, the detection accuracy for reentrancy vulnerability is 96.66%, and the F1-score is 96.74%. For both integer overflow and integer underflow vulnerabilities, the evaluation metrics are around 85%. One possible reason is that reentrancy, timestamp, infinite loop, and callstack depth attack have more salient syntactic or semantic features compared to samples of integer overflow vulnerabilities. SPCBIG-EC also demonstrated good detection performance when performing detection tasks on UCESC datasets containing hybrid vulnerabilities. As the UCESC dataset contains multiple vulnerabilities, its ability to learn vulnerability features decreases when simultaneous feature extraction is performed for multiple vulnerabilities. As a result, the accuracy and F1-scores only reach about 85%.

In summary, the experiments show that SPCBIG-EC can automatically learn semantic and syntactic information by processing and analyzing the source code of smart contracts. Therefore, many different types of vulnerabilities can be detected efficiently. Moreover, our model exhibited good performance in most vulnerability detection tasks. Figure 9 shows the acc-loss plots of our model for different vulnerabilities during the training process. The stability of our model during the training of smart contracts for vulnerability detection can be seen from the accuracy-loss images obtained from the tests during training.

(2) Results for Answering RQ2: To answer the question of whether adopting a serial combination structure of SPCNN and BiGRU in the feature extraction phase makes the SPCBIG-EC model more effective, we conducted experiments from two perspectives.

We compared the performance of vulnerability detection using the SPCNN model, the BiGRU model, and the SPCBIG-EC model to further illustrate the effectiveness of the serial structures based on CNN and RNN.

As shown in Table 4, for the three typical vulnerabilities, our serial structure outperformed CNN or RNN alone in terms of detection performance. (a) In reentrancy vulnerability detection, the evaluation metrics of the models were both improved by about 10% after serializing the CNN with the RNN for feature extraction. Furthermore, SPCBIG-EC achieved the highest accuracy rate of 96.66% with the highest F1-score of 96.74%. (b) In timestamp dependency vulnerability detection, accuracy improved by around 10%, and the F1-score increased by up to 15%. (c) There was a 15% improvement in accuracy and up to 20% improvement in the F1-score in infinite loop vulnerability detection.

The experimental results show that the vulnerability detection capability of the model is greatly improved by concatenating CNN with RNN. The results demonstrate that serially connecting CNNs to RNNs is more conducive to feature extraction and can improve the effectiveness of our model.

We performed hybrid vulnerability detection on UCESC. The performance of all combined models was evaluated using the SCR Detection Framework. Table 5 shows the results of the experiment.

From the results data in Table 5, we can see that the detection performance of the convolutional layer with the SPCNN structure is better than that of CNN and SCNN, with the guarantee of a consistent recurrent neural network. Furthermore, among these 15 combined models, SPCBIG-EC shows stronger vulnerability detection ability. In this experiment, we conducted a comparison of our own structures for vulnerability detection on the same dataset to determine the best model structure, and the experimental results show that the SPCNN structure has a strong advantage over CNN and SCNN; our proposed SPCNN has a factual basis for improving vulnerability detection performance, and our model also shows the strongest vulnerability detection capability among these 15 structures.

(3) Results for Answering RQ3: To answer whether our proposed SPCNN structure can make our model more effective, we computed experimental data statistics with three vulnerabilities as examples. We chose the BiGRU model in the RNN module to detect the performance of the structure by changing the type of CNN.

Table 6 shows the quantitative results, and we have the following observations. (a) In reentrancy vulnerability detection, SPCBIG-EC has an accuracy and F1-score of 96%, much higher than those of CNN-BiGRU and SCNN-BiGRU. The various indicators improved by around 10%. The difference in performance between CNN-BiGRU and SCNN-BiGRU is not significant, and the accuracy and F1-score of SCNN-BiGRU are slightly better than those of CNN-BiGRU. (b) In timestamp vulnerability detection, SPCBIG-EC achieved an accuracy and F1-score of around 91%, slightly higher than those of BiGRU and SPCNN. Although the advantages are not obvious, good vulnerability detection was achieved. (c) In infinite loop vulnerability detection, SPCBIG-EC achieved an accuracy of 94.87% and an F1-score of 95%, much higher those than CNN-BiGRU and SCNN-BiGRU. These metrics are in the range of 5–15% improvement compared to CNN-BiGRU and SCNN-BiGRU metrics.

The experiments showed that the results of vulnerability detection vary greatly when different CNNs are chosen for sequence structure. The results demonstrate that SPCNN can improve the effectiveness of our model, especially the overall effectiveness of the accuracy and F1-score. In addition, we found that the detection performance of reentrancy and infinite loop vulnerabilities improved to a greater extent than that of timestamp vulnerabilities. The possible reason is that the reentrancy and infinite loop samples have a richer data flow and control flow relationship, so feature extraction with SPCNN leads to a greater improvement in the final detection results.

We further present a case study in Figure 10, where the withdraw function is a real-world smart contract function that has a reentrancy vulnerability. We use the reentrancy vulnerability as an example and present the labels of the detection results of the CNN-BiGRU and SPCBIG-EC models. A vulnerability contract was located, and SPCNN detected the vulnerability, while CNN did not detect the vulnerability. We reduced the contract represented by tokens to the smart contract source code. We found that CNN failed to locate the vulnerability in this contract. The vulnerability in this code is in “*require(msg.sender.call.value(_weiToWithdraw)())*”. This line will send the extractor the specified number of Ether. Based on this experimental phenomenon, we found that CNN does not adequately learn and extract the logic of changing state variables in the contract code. In contrast, the SPCNN learns sufficiently, based on semantic analysis and feature extraction, that it is logically appropriate for the contract to change the state variables before Ethereum is invoked externally. This case study further validates the superiority of the SPCNN structure for grammar analysis and feature extraction.

(4) Results for Answering RQ4: To assess the value of our model, we selected state-of-the-art methods for comparison. Firstly, a comparison was made with tools such as Oyenete [25], Mythril [26], Smartcheck [27], and Securify [28] for reentrancy and timestamp dependency vulnerabilities. Secondly, we compared it with deep learning-based vulnerability detection methods, namely, Eth2Vec [33], DR-GCN, TMP, GCE, AME [35], AFS [36], and DeeSCV [38]. These methods are described in detail in Section 2.

To highlight the need for research on smart contract vulnerability detection methods based on deep learning techniques, we compared the performance of traditional smart contract vulnerability detection tools with our approach. Table 7 shows the quantitative results, and we have the following observations. Among the four traditional methods without the deep learning phase, (a) Securify received the highest F1-score in reentrancy vulnerability detection, with a value of 52.79%. (b) In the timestamp dependency vulnerability, Mythril obtained the highest F1-score with a value of 42.48%, which is quite low in practice. This stems from the fact that these methods detect these two kinds of vulnerabilities by crudely checking whether the statements contain *call.value*/*block.timestamp* or not.

To verify the efficiency of our approach, we compared the performance with existing advanced smart contract vulnerability detection methods based on deep learning techniques, and Table 8 shows the resulting data. Figure 11 shows the visualization of our approach compared to existing advanced deep learning-based methods. We analyzed the data from the experimental results. (a) In reentrancy vulnerability detection, AME reached the highest F1-score with a value of 87.94%, and Eth2Vec obtained an F1-score of 61.5%, the lowest value of all methods; (b) For timestamp dependency, CGE reached the highest F1-score with a value of 87.75%, and DR-GCN obtained an F1-score of 74.91%, the lowest value of all methods; (c) In the infinite loop vulnerability, CGE achieved the highest F1-score with a value of 82.13%, and DR-GCN obtained an F1-score of 66.32%, the lowest value of all methods.

In contrast, our method is considerably superior to the state-of-the-art methods described above. Our method obtained a 96.66% F1-score in detecting reentrancy vulnerabilities, a 91.62% F1-score in timestamp dependency, and a 95% F1-score in infinite loop vulnerabilities. These pieces of evidence reveal the great potential of applying our model to smart contract vulnerability detection.

We used multiple tools to perform different types of vulnerability detection on thousands of real-world copies. We selected several typical traditional smart contract vulnerability detection tools to compare the detection time with our method in order to further illustrate the shortcomings of the traditional method and the urgent need to improve it. At the same time, to verify the efficiency of our method, we compared the vulnerability detection time with CGE, the vulnerability detection method based on deep learning techniques with the highest comprehensive performance except for our method in Table 8. Statistically, as shown in Figure 12, our model detected an average of 9.8 s for one smart contract. Meanwhile, Mythril averaged 60 s, Oyente averaged about 30 s, Securify averaged about 20 s, and CGE averaged about 12.4 s. This demonstrates the efficiency of our SPCBIG-EC.

## 5. Analysis of Results and Outlook for Future Work

From the above experimental results, it can be concluded that our model shows good performance when targeting a single vulnerability. This conclusion can be drawn from Table 3 and Table 8. We also find that the types of vulnerabilities where our model performs outstandingly are reentrancy vulnerabilities, timestamp-dependent vulnerabilities, and infinite loop vulnerabilities; especially for reentrant vulnerabilities, we achieved almost 97% detection accuracy and F1-score. On the other hand, when we used our model for hybrid vulnerability detection, we found that our performance metrics only reached 85.89%. The possible reason is that the low accuracy of our model for detecting the three remaining types of vulnerabilities in our study led to low performance metrics when performing hybrid vulnerability detection. To test this hypothesis and to consider the effect of the dataset on the experimental results under equivalent conditions, we performed a comparison of vulnerability detection under equivalent conditions by using the dataset obtained from [39]. The experimental results are shown in Table 9.

Figure 13 shows the accuracy-loss images when using our model for hybrid vulnerability detection on the dataset of [39]. The experimental results data show that the detection results of the same model are different on different datasets. We used the dataset of [39] for hybrid vulnerability detection experiments, and the comparison shows that our model improved the accuracy and F1-score by 6% and the recall by 10% on their dataset. Our hybrid vulnerability detection performance improved to over 90%, outperforming most existing methods. The possible reason for our hybrid vulnerability detection performance not reaching higher values is the bias of our model in its ability to detect different vulnerabilities. We found experimentally that our model obtained poor evaluation metrics for integer overflow vulnerability detection, a conclusion that can be drawn from Table 3. We want to show that our model focuses on targeted vulnerability detection for several single vulnerability characteristics, and that our approach has advantages over existing state-of-the-art approaches for single vulnerability detection for reentry vulnerabilities, timestamp-dependent vulnerabilities, and infinite loop vulnerabilities. Since the dataset of [39] is not divided into different types of vulnerability categories, we cannot compare the performance of a single vulnerability detection with them in an equivalent situation. We acknowledge that their study has a strong detection capability for mixed vulnerabilities. However, our advantage is that we have stronger detection performance for vulnerabilities that have more salient syntactic or semantic features, such as smart contract reentrancy vulnerabilities. The superior detection performance of our model for these types of vulnerabilities is due to the model’s unique string-parallel convolutional structure (SPCNN) and integrated classification algorithm. We use the smart contract reentry vulnerability as an example to illustrate. Considering the correlation characteristics of inter-invocation between smart contracts, the smart contract vulnerability features are extracted and identified by using a string-parallel convolutional structure with parallel dual convolutional kernels, and the internal features of smart contracts and the correlation features between multiple smart contracts are fully extracted.

The experimental comparison of the replacement dataset and the related experiments in Section 4 lead us to the following conclusions. The strength of our model is its strong detection capability against smart contract reentrancy vulnerabilities, which stems from our unique cascading model structure and integrated classification algorithm. In addition, our model has strong detection capability against timestamp dependency vulnerabilities, infinite loop vulnerabilities, and callstack depth attack vulnerabilities, which is better than most similar detection models. In contrast, our model is weak in identifying integer overflow vulnerabilities, which is the reason why our detection ability is lower than that of [39] when performing hybrid vulnerability detection.

The possible reasons why our model is less sensitive to integer overflow vulnerabilities are as follows. The integer overflow vulnerability is mainly due to the integer type of Solidity, the programming language of smart contracts. Both uint8 and uint256 can only store integers within a certain range, and when the result of the operation exceeds that range, an overflow or underflow problem will occur, which makes certain judgment mechanisms in the contract fail and poses a potential threat to the security of the entire contract. Most of the integer overflow vulnerability attacks bypass the detection statements in the contract by carefully constructing parameters to achieve an over-transfer. We believe that our model lacks experience in analyzing the parameters at the stage where the user makes the transfer. In our future research, we will continue to optimize our model to make our approach more capable of analyzing and identifying multiple vulnerabilities such as integer overflows.

Through this research, we realized that future research on deep learning-based smart contract vulnerability detection should be strengthened in the following aspects. (1) Building a unified and standardized smart contract vulnerability dataset. A breakthrough in deep learning-based smart contract vulnerability detection methods must rely on a unified and comprehensive smart contract vulnerability dataset. Currently, the existing deep learning methods (e.g., ReChecker, TMP, and AME) can only support a small number of contract vulnerability detection types because the dataset is poor and non-standardized. Therefore, only a unified and standardized vulnerability dataset covering all types of vulnerabilities can enable deep learning models to have a better effect and thus better promote research in this area. (2) Establish a unified, scalable deep learning model. With the number of smart contracts, the corresponding type of security vulnerability has become more and more complicated and unpredictable. At present, the vulnerabilities in smart contracts based on deep learning can still establish a model on the type of vulnerabilities found, and whether it can quickly adapt to a new vulnerability type is yet to be studied.

## 6. Conclusions

The devices in the IoT ecosystem are the points of contact with the physical world. IoT devices link to the cloud through some communication media. The data collected by the sensor reaches the cloud through the core network for processing [62]. IoT devices are more deeply embedded in the privacy of people’s lives, and their security issues cannot be ignored. Smart contracts backed by blockchain technology have the potential to solve these problems. While BoT technology is vigorously promoting the development of industrial innovation, its security must also be guaranteed. Thus, it is urgent to propose a more efficient approach to smart contract security detection. Our research focuses on six typical vulnerabilities, namely, reentrancy vulnerability, timestamp dependency, infinite loop, callstack depth attack vulnerability, integer overflow, and integer underflow. In addition, we innovatively propose an SPCNN network structure suitable for serial combinatorial models. This structure aims to extract low-level features and output them in parallel with high-level features. It can fully integrate the multiple features of word vectors at different levels. In order to fully extract the internal features of smart contracts and the connection features between smart contracts, we use SPCNN with dual convolutional kernels. Numerous experiments demonstrated the efficiency of the SPCBIG-EC model and the applicability of the SPCNN to serial hybrid models. In addition, experimental data show that our SPCBIG-EC model significantly outperforms 11 other advanced vulnerability detection methods.

## Figures and Tables

**Figure 1 sensors-22-04621-f001:**
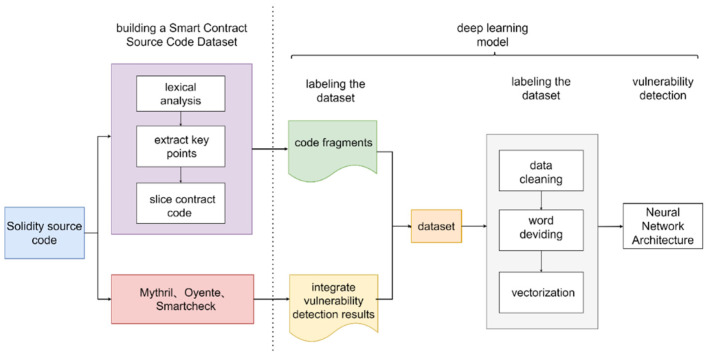
A holistic design solution for smart contract source code vulnerability detection based on deep learning.

**Figure 2 sensors-22-04621-f002:**
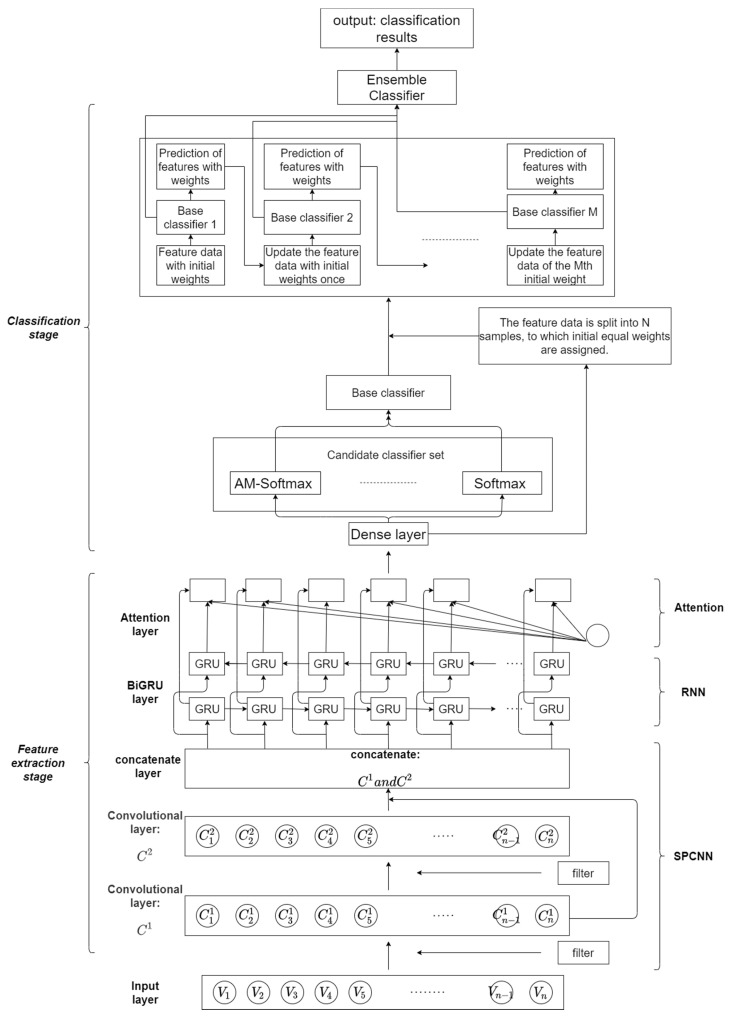
SPCBIG-EC.

**Figure 3 sensors-22-04621-f003:**
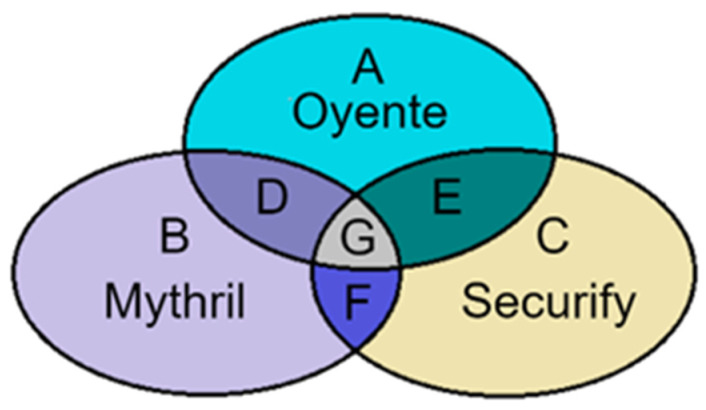
Guidelines for labeling vulnerabilities.

**Figure 4 sensors-22-04621-f004:**
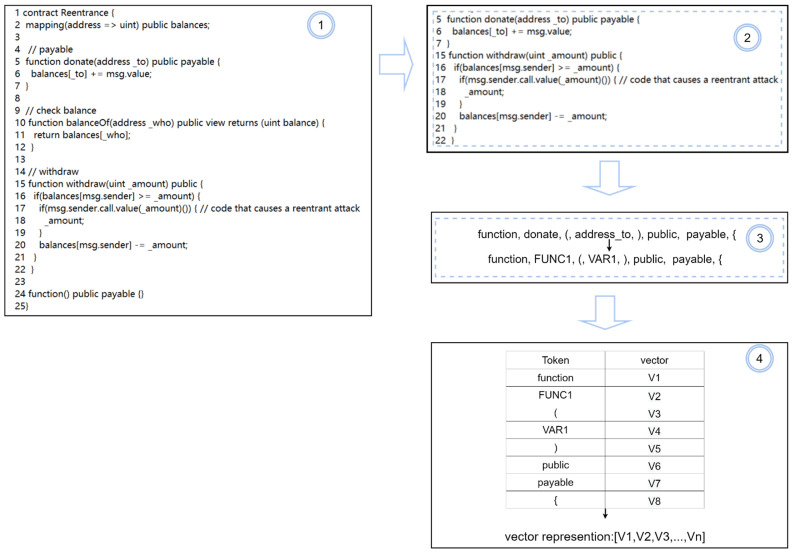
The code embedding process. ①–④ represent the process of representing the original smart contract code into vectors.

**Figure 5 sensors-22-04621-f005:**
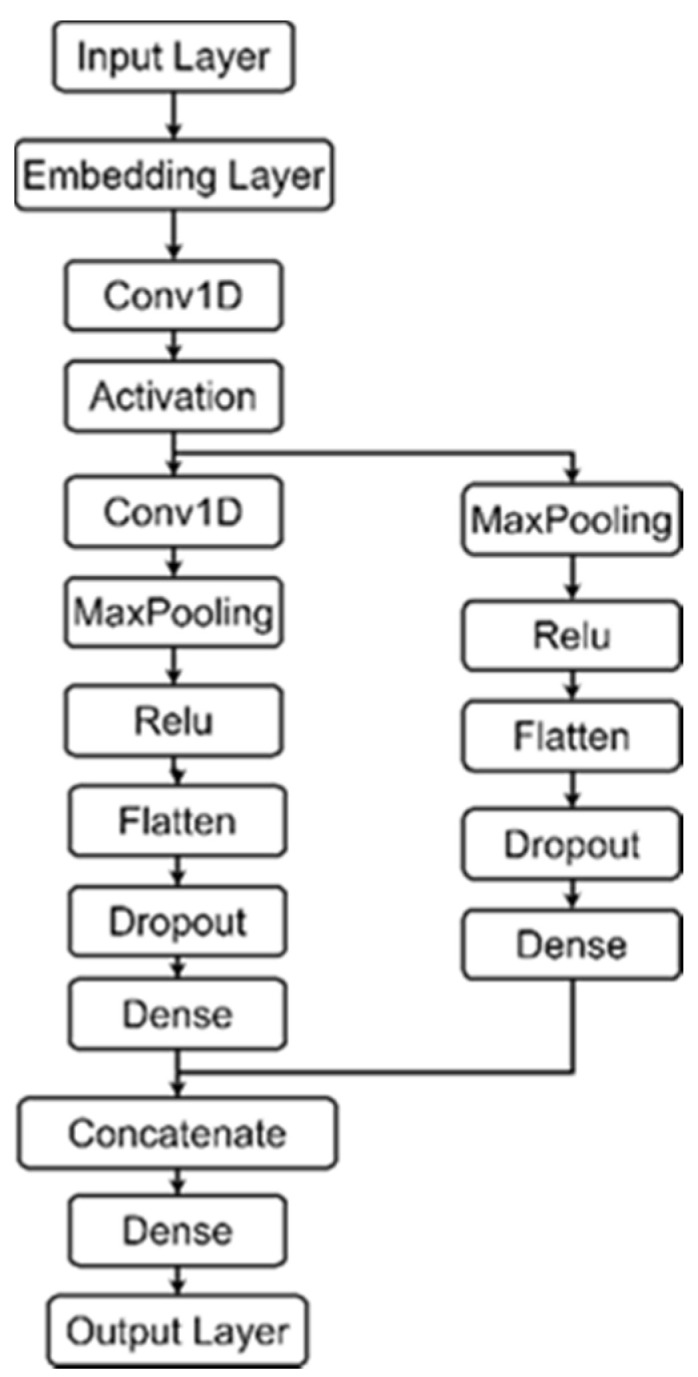
Hierarchy of the SPCNN.

**Figure 6 sensors-22-04621-f006:**
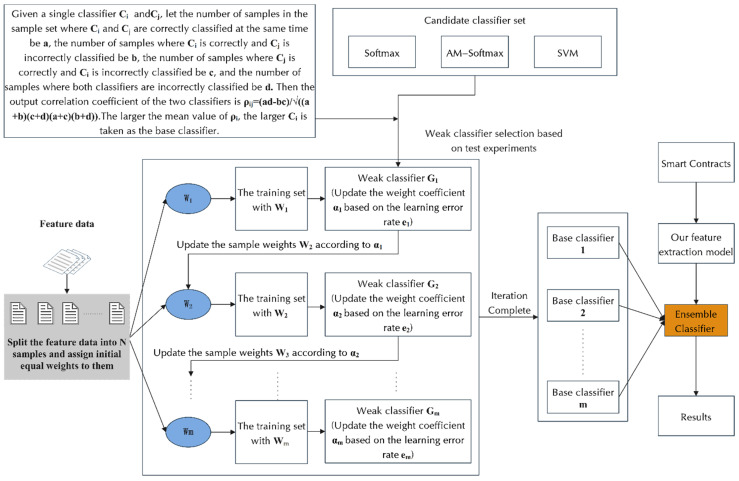
Structure diagram of the Ensemble Classifier.

**Figure 7 sensors-22-04621-f007:**
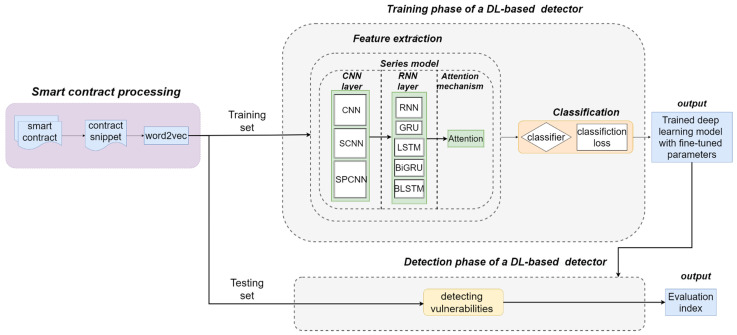
The SCR Detection Framework.

**Figure 8 sensors-22-04621-f008:**
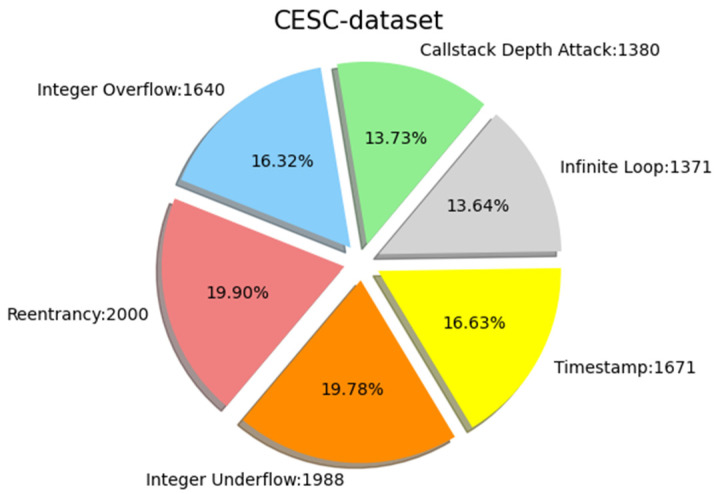
Composition of the CESC dataset.

**Figure 9 sensors-22-04621-f009:**
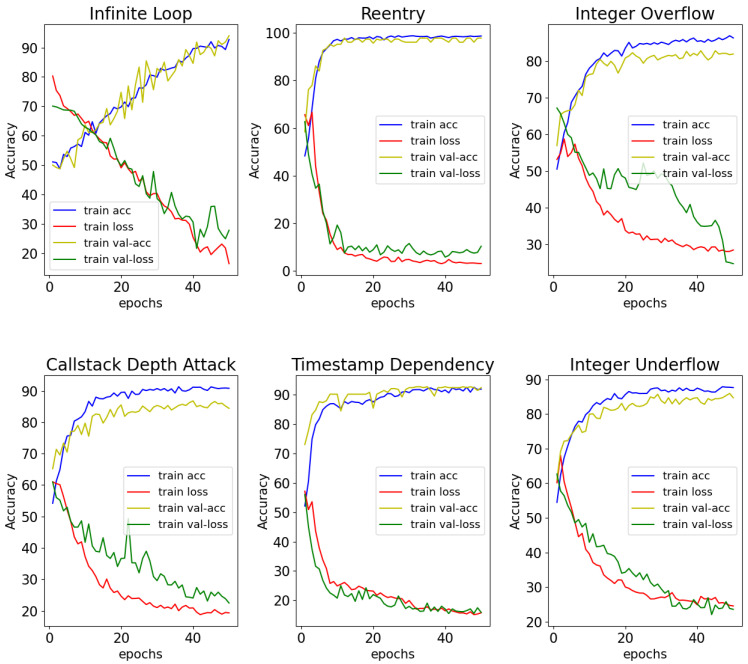
Plots of Acc-Loss.

**Figure 10 sensors-22-04621-f010:**
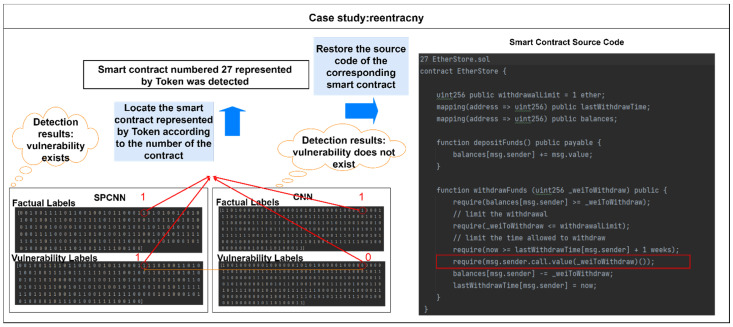
Case study on the interpretability of our method. The “1” and “0” in the graph indicate the result of vulnerability detection. A “1” indicates that a vulnerability exists in the detected contract, and a “0” indicates that it does not exist.

**Figure 11 sensors-22-04621-f011:**
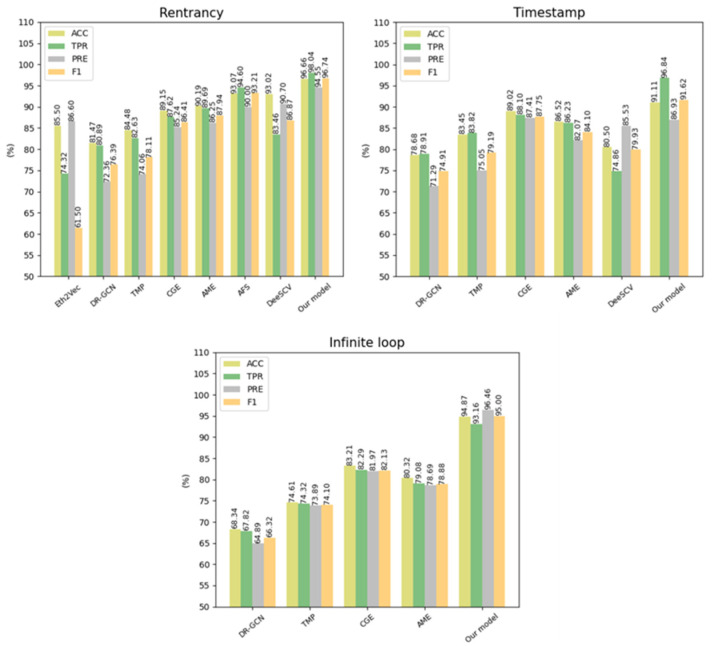
Visualization of images compared to existing deep learning-based methods.

**Figure 12 sensors-22-04621-f012:**
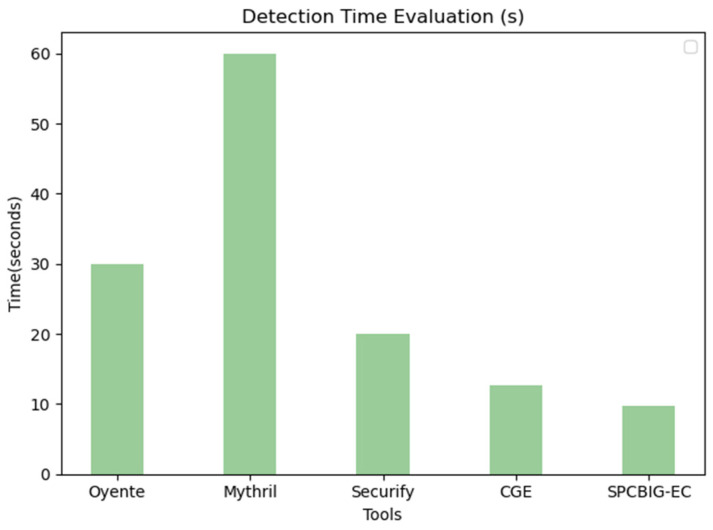
The comparison in detection time.

**Figure 13 sensors-22-04621-f013:**
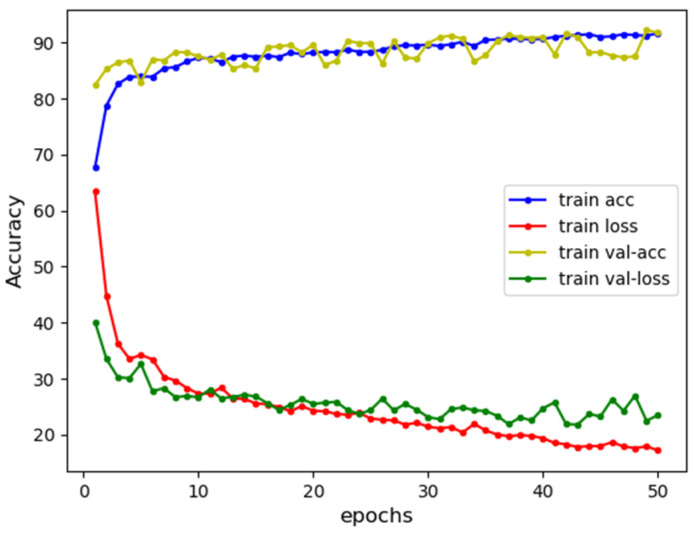
Plots of Acc-Loss during hybrid vulnerability detection.

**Table 1 sensors-22-04621-t001:** The typical Solidity symbol corresponds to the token representation.

Symbolic Representation	Token
function	FUN
contract	CON
for	FOR
if	IF
else	ELSE
number	NUM
constant	CONS
variable	VAR

**Table 2 sensors-22-04621-t002:** The key points corresponding to different vulnerabilities.

Vulnerability	Key Points
Reentrancy	*fallback ()*, *call.value ()*
Timestamp dependence	*block.number*
Infinite loop	*for*, *while loop*
Integer overflow	*integer variables*
Integer underflow	*integer variables*
Callstack depth attack	*.call ()*, *transfer () and the return* *values of instructions*

**Table 3 sensors-22-04621-t003:** The key points corresponding to different vulnerabilities.

Dataset	Vulnerability	Models Selected by Our Framework	ACC (%)	TPR (%)	FPR (%)	PRE (%)	F1 (%)
CESC	Reentrancy	SPCBIG-EC	96.66	98.04	5.71	94.55	96.74
	Timestamp	SPCBIG-EC	91.11	96.84	14.64	86.93	91.62
	Infinite loop	SPCBIG-EC	94.87	93.16	3.41	96.46	95.00
	Callstack depth attack	SPCBIG-EC	90.02	86.32	7.24	92.25	89.14
	Integer overflow	SPCBIG-EC	85.11	97.10	16.90	83.80	85.42
	Integer underflow	SPCBIG-EC	86.47	89.24	16.29	84.57	86.83

**Table 4 sensors-22-04621-t004:** Performance comparison of a single model and serial hybrid model.

	Metrics	ACC (%)	TPR (%)	FPR (%)	PRE (%)	F1 (%)
Model						
Reentrancy	BiGRU	85.38	86.57	11.42	85.23	85.55
	SPCNN	87.14	87.12	12.23	85.45	86.77
	SPCBIG-EC	96.66	98.04	5.71	94.55	96.74
Timestamp	BiGRU	82.45	83.82	13.28	71.29	76.82
	SPCNN	83.48	82.56	12.84	75.27	79.19
	SPCBIG-EC	91.11	96.84	14.64	86.93	91.62
Infinite loop	BiGRU	80.11	83.54	21.45	73.98	76.27
	SPCNN	79.79	85.14	11.84	75.17	78.86
	SPCBIG-EC	94.87	93.16	3.41	96.46	95.00

**Table 5 sensors-22-04621-t005:** The results of the detection of hybrid vulnerabilities with the SCR Detection Framework.

	Metrics	ACC (%)	TPR (%)	FPR (%)	PRE (%)	F1 (%)
Serial Neural Network						
CNN-RNN		75.25	62.56	12.06	83.84	71.65
CNN-LSTM		79.77	79.90	20.35	79.70	79.80
CNN-GRU		80.52	71.11	10.05	87.61	78.50
CNN-BLSTM		80.65	76.63	15.32	83.33	79.85
CNN-BiGRU		81.41	81.91	19.09	81.10	81.50
SCNN-RNN		79.02	72.36	14.32	83.47	77.52
SCNN-LSTM		78.64	82.16	24.87	76.76	79.37
SCNN-GRU		81.40	73.11	10.30	87.65	79.72
SCNN-BLSTM		82.41	72.11	10.29	90.82	80.39
SCNN-BiGRU		82.92	80.90	15.07	84.30	82.56
SPCNN-RNN		79.27	87.69	29.14	75.05	80.88
SPCNN-LSTM		80.15	80.00	19.50	80.30	80.10
SPCNN-GRU		82.03	81.66	17.58	82.28	81.97
SPCNN-BLSTM		82.66	81.16	15.82	83.68	82.40
SPCBIG-EC		85.89	85.00	9.23	91.41	85.71

**Table 6 sensors-22-04621-t006:** The results of the detection of hybrid vulnerabilities with the SCR Detection Framework.

	Metrics	ACC (%)	TPR (%)	FPR (%)	PRE (%)	F1 (%)
Model						
Reentrancy	CNN-BiGRU	84.76	76.19	6.67	91.95	83.33
	SCNN-BiGRU	86.19	93.33	20.95	81.67	87.11
	SPCBIG-EC	96.66	98.04	5.71	94.55	96.74
Timestamp	CNN-BiGRU	87.61	92.72	23.42	80.73	88.83
	SPCNN-BiGRU	90.47	91.77	10.83	89.5	90.62
	SPCBIG-EC	91.11	96.84	14.64	86.93	91.62
Infinite loop	CNN-BiGRU	80.77	72.65	11.11	86.73	79.0
	SPCNN-BiGRU	89.74	88.03	8.55	91.15	89.57
	SPCBIG-EC	94.87	93.16	3.41	96.46	95.00

**Table 7 sensors-22-04621-t007:** Comparison results with advanced security audit tools.

Methods	Reentrancy				Timestamp			
	ACC (%)	TPR (%)	PRE (%)	F1 (%)	ACC (%)	TPR (%)	PRE (%)	F1 (%)
Oyenete	71.50	50.84	51.72	51.28	60.54	31.94	48.94	38.66
Mythril	60.00	39.21	68.96	50.00	64.87	34.53	58.54	42.48
Smartcheck	52.00	24.32	31.03	27.27	61.08	29.17	50.00	36.84
Securify	53.50	37.41	89.66	52.79	-	-	-	-
SPCBIG-EC	96.66	98.04	94.55	96.74	91.11	96.84	86.93	91.62

**Table 8 sensors-22-04621-t008:** Comparison results with existing advanced deep learning-based methods.

Methods	Reentrancy				Timestamp				Infinite Loop			
	ACC (%)	TPR (%)	PRE (%)	F1 (%)	ACC (%)	TPR (%)	PRE (%)	F1 (%)	ACC (%)	TPR (%)	PRE (%)	F1 (%)
GR-GCN	81.47	80.89	72.36	76.39	78.68	78.91	71.29	74.91	68.34	67.82	64.89	66.32
TMP	84.48	82.63	74.06	78.11	83.45	83.82	75.05	79.19	74.61	74.32	73.89	74.10
CGE	89.15	87.62	85.24	86.41	89.02	88.10	87.41	87.75	83.21	82.29	81.97	82.13
AME	90.19	89.69	86.25	87.94	86.52	86.23	82.07	84.10	80.32	79.08	78.69	78.88
Eth2Vec	85.50	74.32	86.60	61.50	-	-	-	-	-	-	-	-
AFS	93.07	94.60	90.00	93.21	-	-	-	-	-	-	-	-
DeeSCV	93.02	83.46	90.70	86.87	80.50	74.86	85.53	79.93	-	-	-	-
SPCBIG-EC	96.66	98.04	94.55	96.74	91.11	96.84	86.93	91.62	94.87	93.16	96.46	86.93

**Table 9 sensors-22-04621-t009:** Comparison of hybrid vulnerabilities under different datasets.

Model	Dataset	ACC (%)	TPR (%)	FPR (%)	PRE (%)	F1 (%)
SPCBIG-EC	UCESC	85.89	85.00	9.23	91.41	85.71
	SCVDIE-Ensemble [39]	91.80	95.25	11.67	90.00	92.10

## Data Availability

The experimental data and associated code used in this study have been deposited in the GitHub repository (https://github.com/wobulijie10086/SPCBIG-EC/tree/master/SPCBIG-EC, accessed on 30 March 2022).

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
