# Peer review of "SPCBIG-EC: A Robust Serial Hybrid Model for Smart Contract Vulnerability Detection"

_sensors, 2022, doi:10.3390/s22124621_

Round 1

Reviewer 1 Report

The problem studied is an interesting one for researchers working in this topic.

  1. The objectives are not clearly mentioned in the early part of the paper. Only after reading the conclusion, I can understand the main objectives. 
  2. Related work should support the objectives.
  3. Figures are hard to read and understand. Font size must be improved.

Reviewer 2 Report

The proposed model achieved better performance results that proposed by other studies.

Motivation for choosing RNN an CNN is provided.

Very detailed methodology that allows to reproduce the experiment and method.

Lines  57-63: all claims about the combination of IoT and smart contract, their benefits and problems need to be supported by referencing to the existing research studies or other sources.

Line 106: Did you mean Etherium platform?

Line 150: Third, they have a long audit time. – compared to what methods? You need to compare the results of your research with other most efficient methods

Figure 2  - difficult to make association between text and Figure. It seems that Figure with model does not include steps 1 and 2 of the model text version. Please provide a parallels for the text and Figure.

Figure 10 – not readable. the quality needs to be improved

Line 670 – comparison analysis, more information about models is required, for example, how they are related to Related works section? Provide references.

The Table 7nincludes the comparison with methods that showed low effectiveness in many studies [24-29]. After these methods many new more effective methods have been proposed.

Same with execution time, new methods need to be considered for comparison.

Reviewer 3 Report

In order to solve the security of smart contracts, this manuscript proposes a flexible and systematic hybrid model for smart contract vulnerability detection. More specifically, the author propose a serial-parallel convolution (SPCNN) suitable for our hybrid model, sensing and communications into account. In SCR Detection Framework, it contains 15 serial combinations of CNN and RNN models including SPCBIG-EC. The main feature of this paper is the construction of deep learning models, in which build serial hybrid models combining CNN and RNN.Finally, simulation results are given.

————————

Strength:

1) Motivation is good and problem is well articulated.

2) Both theoretical and experimental analysis are given.

3) The experiments results seem to be reasonable and practical.

————————

Weakness:

1) The figures in this article are very blurry. Please use a clear original image

2) The author is motivated by the considerations,but The author only expounds the relevant concepts and principles, and does not give the correlation between the method in this paper and this principle.The author should explain the rationales underlying their method, and the limitations of their method.

3) As stated in Fig 1, the deep learning model also needs to label data, but the author separated label data and the deep learning method. It is suggested to redivide the structure of Figure 1.

4) I suggest the authors to explicitly define the threat model, that is what are the capabilities allowed to an attacker. Otherwise, it is difficult to assess the security of the Serial Hybrid Model.
"OPAQUE:An Asymmetric PAKE Protocol Secure Against Pre-Computation Attacks", Proc. EUROCRYPT 2018. 
"Quantum2FA: Efficient Quantum-Resistant Two-Factor Authentication Scheme for Mobile Devices", IEEE Transactions on Dependable and Secure Computing, 2021
"Secure and Usable Handshake based Pairing for Wrist-Worn Smart Devices on Different Users". Mobile Networks and Applications, 2021. DOI: 10.1007/s11036-021-01781-x

5)As stated in Section 3 "Design of the model", there are a number of components  involved in the system. How a  component ensures itself to other components that it is legitimate and is allowed to perform specified (but not malicious) actions? Is there some form of authentication needed? Are the following authentication schemes suitable? The authors may briefly discuss this. 
"Unified Biometric Privacy Preserving Three-factor Authentication and Key Agreement for Cloud-assisted Autonomous Vehicles". IEEE Transactions on Vehicular Technology. DOI: 10.1109/TVT.2020.2971254
"Practical and Provably Secure Three-Factor Authentication Protocol Based on Extended Chaotic-Maps for Mobile Lightweight Devices", IEEE Transactions on Dependable and Secure Computing, 2022, 19(2):1338-1351.
"Evaluating Behavioral Biometrics for Continuous Authentication: Challenges and Metrics", Proc. Asiaccs 2017.

6) “As shown in Table 2.”The introduction of Table2 is not very clear, it is suggested to modify the relevant statements. It is recommended that the formula cited in the paper be identified in References.

————————

In all, I suggest a Major revision.

Reviewer 4 Report

Authors claim:

1. Propose SPCBIG-EC model
2. It has excellent performance (multi-task vulnerability)
3. Propose Serial-Parallel CNN
4. Use ensemble classifier
5. Constructed novel datasets 
6. Outperforms existing models

@1. The model is not clearly explained. Fig 2. should depict the model, but "features" are not explained, "SP-CNN" layer is not explained, and classification stage does not show ensemble learning. Which classifiers are used in ensemble learning? Please also see @3. Similar questions hold for "bidirectional gated" - I cannot relate the name with the important properties of the proposed architecture.

@2. The authors did not convinced me that it has excellent performance (Please see @6). The also use term "multi-task vulnerability". What does it even suppose to mean?

@3. Unless clearly explained what authors mean by serial, and what do they mean under parallel, serial-parallel sounds like an oxymoron. How the architecture is different than CNN?

@4. Ensemble classifier could be reinventing the wheel. Ensemble learning should combine multiple classifiers. I do not see this is done. This seems as a single layer neural network similar to ADALINE with Heaviside step function as a nonlinearity.

@5. The authors claim that they construct novel datasets. I do not see them available, and this contribution is perhaps irrelevant for the average reader of the journal but perhaps more suitable for a "data" journal.

@6. The authors claim their model outperforms existing models with acc and F1 > 85%. This is obviously false claim. Please see recently published paper by Zhang et al. "A Novel Smart Contract Vulnerability Detection Method Based on Information Graph and Ensemble Learning"

Round 2

Reviewer 3 Report

The revised version has addressed most of my concerns, and I suggest an acceptance.

Author Response

Thank you very much for your comments and suggestions. We have made changes in accordance with your suggestions and improved the language presentation.

Reviewer 4 Report

@1 - Thank you for your explanation

@2 - I agree that tables show that SPCBIG-EC performs better than the other models shown in the tables. We do not agree that the model outperforms existing methods.
True, if we decide to compare just the methods from the table, and discard other research we can state that the method outperforms other methods.

@3 - When authors describe serial parallel architecture, they are describing skip layer architecture. There is nothing novel about this.
Moreover, they sometimes use the term "string parallel" for SP.

@4 - I'm still not satisfied with the explanation of ensemble learning. Which classifiers are used as a weak classifiers? How many of them? How are they combined? Is there any previous work on why this type of "ensemble learning" is better than classical approach?

@5 - So, we agree that the construction of the novel dataset is not relevant novelty for this manuscript.

@6 - same as @2 - we can agree that the paper outperforms some methods, but we should also agree that there are other, already published, methods that outperform the results provided in this manuscript. Detection is (by definition) 0/1 classification problem (True/False). I do not see how these two papers do not fall into the same category. Perhaps authors are not solving the detection problem... Vulnerability detection does not try to determine the type of vulnerability. Finding the type of vulnerability is the classification problem.

From @3 & @4 I must conclude that there is nothing novel in the methodology.
From @5 follows that we agree that the dataset is not the main focus of this paper.
From @2 & @6 follows that if we exclude the methods that have higher detection rate than SPCBIG-EC, the SPCBIG-EC outperforms all others method - which is tautology. It also seems that authors are not aiming to solve the detection problem, which should suggest that the title and the experimental setup is inappropriate.
Therefore, I do not find sufficient arguments to suggest this manuscript to be published.
